# Machine Vision-Based Real-Time Monitoring of Bridge Incremental Launching Method

**DOI:** 10.3390/s24227385

**Published:** 2024-11-20

**Authors:** Haibo Xie, Qianyu Liao, Lei Liao, Yanghang Qiu

**Affiliations:** School of Civil Engineering, Changsha University of Science and Technology, Changsha 410004, China; bridgexhb@126.com (H.X.); l13027300037@126.com (L.L.); 15974290924@163.com (Y.Q.)

**Keywords:** displacement monitoring, machine vision measurement, incremental launching construction, multi-target detecting and tracking

## Abstract

With the wide application of the incremental launching method in bridges, the demand for real-time monitoring of launching displacement during bridge incremental launching construction has emerged. In this paper, we propose a machine vision-based real-time monitoring method for the forward displacement and lateral offset of bridge incremental launching in which the linear shape of the bottom surface of the girder is a straight line. The method designs a kind of cross target, and realizes efficient detection, recognition, and tracking of multiple targets during the dynamic process of beam incremental launching by training a YOLOv5 target detection model and a DeepSORT multi-target tracking model. Then, based on the convex packet detection and K-means clustering algorithm, the pixel coordinates of the center point of each target are calculated, and the position change of the beam is monitored according to the change in the center-point coordinates of the targets. The feasibility and effectiveness of the proposed method are verified by comparing the accuracy of the total station and the method through laboratory simulation tests and on-site real-bridge testing.

## 1. Introduction

The incremental launching construction of bridges is characterized by minimal disturbance to existing traffic and the surroundings, strong spanning capacity, and a high amount of formwork turnover. It has been widely applied in the construction of bridges crossing railways, rivers, and other obstacles worldwide [1,2,3,4]. For example, the Jinan Huanghe Bridge in China, a continuous steel truss girder bridge with suspension stiffening chords, was constructed by incremental launching [5]. The Iowa River Bridge in the United States used a relatively unique incremental submersion method for a steel I-beam bridge [6]. The Park Bridge and the Coast Meridian Overpass in Colombia were both erected using specialized incremental launching techniques [7]. Similarly, The Pavilion Bridge in Spain, which is a complex structure of a hybrid pavilion–bridge structure, was constructed using the incremental launching method and continuous monitoring of structural behavior through cross-checking data [8].

The incremental launching method is widely used worldwide. However, during the process of incremental launching construction, the beam structure is susceptible to deviation from the designed axial position due to a variety of factors; therefore, it is necessary to perform real-time monitoring, correction, and prediction of beam structure positions to ensure that they are maximally coincident with the designed axial line [9]. The traditional method generally adopts total stations for monitoring. Zhao et al. [10] used total stations for monitoring during incremental launching construction, and when significant deviation occurred, it was manually adjusted using jackscrews. The total station instrument obtains the three-dimensional measured coordinates of the control points to calculate the positional changes of the beam structure, but this method is prone to measurement interference and inefficiency in measurement and cannot provide real-time feedback. With the continuous progress of science and technology, Global Navigation Satellite System (GNSS) technology is characterized by a high sampling rate, all-weather monitoring, and a high degree of automation, which shows advantages in the field of bridge structural health monitoring (SHM) [11,12,13]. Kashima et al. [14] installed a GNSS monitoring system to measure the deformation of the girders and tower of the Akashi Kaikyo Bridge as early as 1998 and accurately acquired lateral displacements with a vibration amplitude of 0.78 m. Ashkenazi et al. [15] carried out GNSS monitoring experiments on the Humber Bridge in the UK to verify the feasibility of the application of GNSS technology to the dynamic monitoring of bridges. Launching of Iowa River Bridge Pier 4 and launching of the first girder pair for the Park Bridge as shown in Figure 1.

For bridge structures in GPS-limited environments, our goal is to provide continuous, high-precision, and intelligent monitoring during construction. Machine vision, with its unique advantages of non-contact measurement, high precision, and multi-point synchronous acquisition, has garnered increasing attention across various fields in civil engineering in recent years [16]. For example, Yongding Tian et al. [17] utilized an unmanned aerial vehicle (UAV) and computer vision to conduct non-contact cable force measurement. Y Cheng et al. [18] employed machine vision to monitor the assembly of prefabricated structures and validated the accuracy and effectiveness of this method on actual bridges. Shang Jiang et al. [19] proposed using a wall-climbing unmanned aerial system (UAS) to create a crack image database and deployed the trained model into an Android application enabling real-time crack detection on a smartphone.

Meanwhile, the applications of machine vision in intelligent bridge monitoring are gradually increasing. For example, Xing Lei et al. [20] proposed a Scheimpflug camera-based method for multi-point displacement monitoring of bridges and validated its effectiveness through three experiments. Xin Duan et al. [21] analyzed the relative positional changes of natural texture feature points on bridge surfaces before and after deformation, proposed a displacement field calculation theory, and established a full-field displacement monitoring method for structures with natural textures. Yuequan Bao et al. [22] combined machine vision and deep learning for high-precision anomaly detection in bridge structural health monitoring. Adam Marchewka et al. [23] proposed the use of UAV remote sensing technology and digital image processing to realize real-time monitoring of steel bridges. Billie F. Spencer Jr. et al. [24] summarized the research on machine vision for the automated inspection and monitoring of civil infrastructure. Yan Xu et al. [25] summarized the key work in the field of vision-based structural displacement monitoring systems. B. Conde et al. [26] proposed a novel approach using an inverse analysis procedure to investigate pathological issues in masonry arch bridges and conducted experimental validation of the method’s feasibility on the Kakodiki Bridge. Tomasz Garbowski et al. [27] proposed a procedure based on dynamic tests supplemented with several static measurements to determine the largest number of parameters as possible in a short time within an inverse analysis approach, thereby providing a comprehensive method for structural diagnostics of bridges. Xi Liu et al. [28] developed a high-precision surrogate model based on deep learning for the rapid inverse analysis of concrete arch dams. The proposed method was tested on an actual ultra-high concrete arch dam and achieved a 95.83% increase in computational efficiency compared to the direct finite element method.

However, research on the application of machine vision technology for displacement measurement during bridge incremental launching has been rarely reported. This paper proposes a real-time monitoring method for the bridge incremental launching process based on machine vision. It first addresses the challenges of complex multi-point target tracking during girder incremental launching by introducing the YOLOv5 target detection model [29] and DeepSORT multi-target tracking model [30]. The extracted target areas are then processed to obtain the center-point coordinates, which represent the displacement changes during the bridge incremental launching construction according to the sequential position changes of the targets.

## 2. Principles of Machine Vision Measurement

### 2.1. Camera Models

The camera model is a simplification of the optical imaging model, involving four coordinate systems: the world coordinate system, the camera coordinate system, the image coordinate system, and the pixel coordinate system. It also includes their transformation, which maps spatial points of the photographed object to the corresponding points in the image. It is schematically illustrated in Figure 2.

### 2.2. Camera Calibration

Camera calibration is the process of determining the internal and external parameters of the camera, with the internal parameters referring to the camera’s focal length, the position of the principal point, the coefficient of aberration, etc., and the external parameters referring to the camera’s position and attitude (translation vectors and rotation matrices). This process aims to determine the relationship between image coordinates and real 3D coordinates so as to accurately realize the mutual conversion from 2D coordinates to spatial 3D coordinates. In this paper, the method used is Zhang’s calibration [31] and camera calibration using MATLAB2024 as shown in Figure 3. Then obtains the internal parameters of the camera, $f_{x}$ = 2459.4, $f_{y}$ = 2467.9, $c_{x}$ = 1995.0, and $c_{y}$ = 1504.3, and the external parameters, $k_{1}$ = 0.1692, $k_{2}$ = 0.8076, $k_{3}$ = 1.2050, $p_{1}$ = 0.0020, and $p_{2}$ = 0.00012.

## 3. Bridge Incremental Launch Construction Displacement Monitoring System

Artificial markers can increase the distinction between the tracking target and the background, thereby improving the measurement accuracy [32]. So, the design in this paper adopts a type of cross-crossing marker, as shown in Figure 4. The arrangement of the target markers mainly considers the limitation of the camera field of view in the direction of the longitudinal bridge, as well as the setting of the displacement calculation method. During the beam structure incremental launching, when the camera field of view includes at least one complete target, the distance L between neighboring targets must meet the following conditions:(1)$$L<\frac{D\cdot SW}{f}-2W_{region}$$
where L is the arrangement distance of adjacent targets; D is the distance from the camera to the bottom surface of the beam; SW is the width of the camera sensor; f is the focal length of the lens; and $W_{region}$ is the width of the set transition region. To ensure that the subsequent calculations are all for the intact cross targets, a region is set on both sides of the field-of-view area to filter out the incomplete targets, and the value is set to 100.

A GoPro motion camera is installed on the stabilizing mechanism located at the lower part of the beam. It observes the targets that appear sequentially within the camera’s field of view during the incremental launching process, as illustrated in Figure 5. By disregarding the beam’s deformations, manufacturing errors, and the effects of the beam lifting and lowering during the incremental launching process, the process can be simplified to two-dimensional rigid body motion. Analyzing the displacement of the targets within the camera’s field of view allows for the prediction of motion in specific regions of the beam.

Based on the initial position of the appeared target, its straight-line distance to the next appeared target, the angle of the target at its initial position, and the initial position of the unobserved target can be estimated, as shown in Figure 6. The relationship between the initial positions of neighboring targets can be calculated by the following equation:(2)$$\{\begin{array}{c} x_{n+1}=x_{n}-d\cdot \cos\beta \\ y_{n+1}=y_{n}-d\cdot \sin\beta \end{array}$$
where $x_{n}$ and $y_{n}$ represent the initial position of the n-th appearing target; $x_{n+1}$ and $y_{n+1}$ represent the initial positions of the (n+1)-th appearing target; n≥2; d is the straight-line distance between the n-th target and the (n+1)-th target; and β is the angle between the initial actual central axis of the beam in the pixel coordinate system and the transversal coordinate axis, which is the target angle.

By comparing the center-point positions of the tracked targets with corresponding IDs in the image sequence to their initial center-point positions, the pixel displacement can be obtained. A relative displacement conversion relationship is then established between the designed central axis in the real world and the pixel coordinate system. Using the proportional factor method, the pushing displacement and lateral deviation of the measurement point relative to the designed central axis are calculated. When the angle between the camera optical axis and the normal to the bottom surface of the beam is θ, the scale conversion factor (CF) is calculated using the following formula [33]:(3)$$CF=\frac{d_{mm}}{d_{pixel}\cdot cos^{2}\left(\theta \right)}\;or\;\;CF=\frac{D}{f\cdot cos^{2}\left(\theta \right)}d_{pixel}$$
where $d_{mm\;}$ is the size of the target in the structural plane; $d_{pixel}$ is its corresponding pixel size in the image plane; f is the focal length of the lens; and D is the distance from the camera to the bottom of the beam.

Relative displacement conversion is calculated using the following equation:(4)$$\left[\begin{array}{c} \Delta x\\ \Delta y \end{array}\right]=CF\cdot \left[\begin{array}{cc} \cos\left(\alpha +\beta \right) & -\sin\left(\alpha +\beta \right)\\ \sin\left(\alpha +\beta \right) & \cos\left(\alpha +\beta \right) \end{array}\right]\left[\begin{array}{c} \Delta u\\ \Delta v \end{array}\right]$$
(5)$$\left[\begin{array}{c} \Delta u\\ \Delta v \end{array}\right]=\left[\begin{array}{c} x_{i}-x_{0}\\ y_{i}-y_{0} \end{array}\right]$$
where Δx is the incremental launching displacement of a target position of the beam with respect to the designed central axis; Δy is the lateral deflection of a target position of the beam with respect to the designed central axis; Δu and Δv are the horizontal and vertical displacements of a target in the pixel coordinate system, respectively; $\left(x_{0}\;,\;y_{0}\right)$ is the initial position of a target; $\left(x_{i}\;,\;y_{i}\right)$ is the target’s position in the subsequent image sequence; α is the angle between the actual initial central axis of the beam and the designed central axis; and β is the angle of the actual initial central axis of the beam in the pixel coordinate system. The direction of rotation of the angle is determined by the vector cross- product, and the counterclockwise direction is positive.

### 3.1. Based on YOLOv5 Target Detection

YOLOv5 (you only look once) is a single-stage target detection algorithm that enables fast, end-to-end prediction to identify multiple targets in an input image. It consists of four main components: the Input, Backbone, Neck, and Head networks. The Input provides effective pre-processing, which includes Mosaic data enhancement, resizing the input image to a fixed size, and adaptively calculating the size and position of the anchor frame. The Backbone network is primarily composed of modules such as CBS, C3, and SPPF. The C3 module improves the feature extraction capabilities by increasing the network’s depth and receptive field. Fast Spatial Pyramid Pooling (SPPF) is an improved version of SPP that enables faster region pooling operations, thereby enhancing the model’s speed and accuracy when handling inputs of varying sizes. The Neck network performs multi-scale feature fusion on the feature map, while the Head network is responsible for the final regression prediction, obtaining class and location information of the targets and eliminating overlapping bounding boxes using non-maximal suppression methods. The YOLOv5s network structure used in this paper is shown in Figure 7.

The performance of a deep learning model depends directly on the dataset it is trained on. In this paper, a self-collected dataset is used to augment the cross targets with data, which not only increases the number of data samples but also makes the model more robust to handle different input variations and improves its performance in real-world applications. The dataset of this paper has a total of 6490 images, which are divided into 4543 for the training set, 1298 for the validation set, and 649 for the test set according to the ratio of 7:2:1 that was used to divide the dataset. 

Partial data enhancement effect diagram as shown in Figure 8.

The YOLOv5 model was trained on the training set using a pre-trained model. The network profiles chosen were YOLOv5s, YOLOv5s6, YOLOv5n, and YOLOv5n6. The training consisted of 300 epochs with a batch size of 16. Stochastic Gradient Descent (SGD) was used as the optimizer, with an initial learning rate of 0.01 and cosine decay applied. The optimal weights were preserved after several training sessions.

The Average Precision (AP) represents the area under the precision–recall curve for each category, indicating the model’s performance on a specific class. The mean Average Precision (mAP) is obtained by averaging the AP values across all categories, providing a measure of overall performance. The mean Average Precision (mAP) is calculated using the following formula:(6)$$\{\begin{array}{c} AP=\int_{0}^{1}P_{i}\left(R_{i}\right)dR\\ mAP=\frac{\sum_{i=1}^{N_{c}}\int_{0}^{1}P_{i}\left(R_{i}\right)dR}{N_{c}} \end{array}$$

The experimental results comparing YOLOv5s, YOLOv5s6, YOLOv5n, and YOLOv5n6 of the YOLOv5 series of algorithms on the target detection test set are shown in Table 1. As seen in Table 1, all four models have an mAP above 91% and have high detection accuracy. YOLOv5n has the highest precision of 99.2%, its model parameter count is only 56% of that of YOLOv5n6, and its detection time is only 32.8% of that of YOLOv5n6. These results show that YOLOv5n balances the detection accuracy and detection speed, providing fast and accurate detection results for DeepSORT. The results of the crosshatch target detection are shown in Figure 9.

### 3.2. Based on DeepSORT Target Continuous Tracking

The DeepSORT algorithm uses the Re-identification (ReID) algorithm to extract the apparent features of the target and then constructs a cost function through the cosine distance to calculate the similarity between the predicted object and the detected object, which allows for recovering the target’s ID even when it is completely occluded and subsequently reappears. The feature extraction model is chosen to be a lightweight Omni-Scale Network (Omni-Scale Network, OSNet) for learning full-scale feature descriptions [34], and the weight file used corresponds to osnet_x0_25_imagenet.

To validate the effectiveness of the cross-marked target multi-target tracking method, multiple videos that include target occlusion, disappearance, and reappearance in real scenes are selected as the test set. The results of cross-marked target tracking are illustrated in Figure 10, where the number in the upper left corner of each target box represents the ID assigned to that target.

### 3.3. Precise Positioning of the Center-Point Coordinates of the Cross Target

After obtaining the initial position information and ID information of the target through the trained YOLOv5 target detection model and DeepSORT multi-target tracking model, it is necessary to further process the image features of the ROI to calculate the center coordinates of the target, as shown in Figure 11 and Figure 12.

## 4. Experimental Validation and Results

### 4.1. Simulation Experiment

To preliminarily verify the accuracy of the proposed method under ideal conditions, simulation tests were conducted to evaluate its performance by comparing the results with those from total station measurements. A simple platform was constructed by using a herringbone ladder to carry a wooden board, the camera measurement points and total station measurement points were arranged on the side and bottom of the board, respectively, and two reference measurement points were set on the wall; the test arrangement is shown in Figure 13. At the beginning of the test, the movement of the plank was captured using a GoPro 11 motion camera at a frame rate of 30 fps with a resolution of 5312 × 2988. The plank motion was carried out by manual traction, and for each movement, measurements were taken using a Leica TS60 total station to calculate the change in position of the measurement point relative to the reference measurement point.

The displacement calculation of the measurement point was obtained by comparing the position change between two total station reference points A $\left(x_{1},\;y_{1}\right)$ and B $\left(x_{2},\;y_{2}\right)$ and the initial moment point C $\left(x_{i},\;y_{i}\right)$ and the subsequent point D $\left(x_{j},\;y_{j}\right)$ at a certain moment in time, as shown in Figure 14 and Equations (7) and (8).
(7)$$\{\begin{array}{c} \vec{v}=\left(x_{2}-x_{1},y_{2}-y_{1}\right)\\ \vec{u_{0}}=\left(x_{i}-x_{1},y_{i}-y_{1}\right)\\ \vec{u_{1}}=\left(x_{j}-x_{1},y_{j}-y_{1}\right) \end{array}$$
(8)$$\{\begin{array}{c} Lateral\;Offset=h_{2}-h_{1}=\frac{\vec{v}\times \vec{u_{1}}}{\left\|\vec{v}\right\|}-\frac{\vec{v}\times \vec{u_{0}}}{\left\|\vec{v}\right\|}\\ Forward\;Displacement=d_{2}-d_{1}=\frac{\vec{v}\cdot \vec{u_{1}}}{\left\|\vec{v}\right\|}-\frac{\vec{v}\cdot \vec{u_{0}}}{\left\|\vec{v}\right\|} \end{array}$$

Considering the close distance between the setup measurement points and only weak differences in the position change trend, only the displacement comparison results of measurement point 1, shown in Table 2 and Figure 15, and the visual measurement results in Figure 16 are shown.

Due to the randomness of manual pulling, the consistency of each movement cannot be guaranteed, and the lateral deviation fluctuates greatly. The operation time of each traction is short, and the visual measurement results show a stepped trend, with a sudden change in displacement for each movement, and the plateaued section is the time period of the total station measurement after each movement. The test error was measured by Normalized Root Mean Squared Error (NRMSE) with the following formula:(9)$$NRMSE=100\%\times \frac{\sqrt{\frac{1}{n}\sum_{i=1}^{n}{\left(x_{i}-y_{i}\right)}^{2}}}{y_{max}-y_{min}}$$
where n is the number of measured data; $x_{i}$ is the data measured by the visual method; yi is the data measured by the total station; and $y_{max}$ and $y_{min}$ are the maximum and minimum values of the data measured by the total station, respectively.

An analysis of the video of the target in a stationary state was conducted to compare the variations in the target center position over the time series. The standard deviation of the X-coordinate was 0.3417 pixels, with a variance of 0.1167 pixels, while the standard deviation of the Y-coordinate was 0.3413 pixels, with a variance of 0.1165 pixels. The standard deviation and variance of both coordinates indicate that the proposed method has a certain degree of stability. Table 2 lists the NRMSE of this paper’s method for monitoring the displacements of different measurement points. From Table 3 and Figure 15, it can be seen that the calculation results of the two methods are highly consistent, and the trends of the two displacements are very similar, which indicates that this paper’s method has a better monitoring effect.

### 4.2. Real Bridge Test

In order to verify the effectiveness of the proposed method in practical situations, it was tested on a real bridge at the construction site of the main bridge incremental launching of Lixizhou Bridge in Ganzhou, Jiangxi Province. The bridge is a steel truss bridge with an orthotropic–anisotropic steel deck structure. The synchronized incremental launching construction process was applied, and the site situation is shown in Figure 17. The field test arrangement is shown in Figure 18. The incremental launching process was recorded using a motion camera with a resolution of 5312 × 2988 and a frame rate of 30 fps, and a surveyor’s app Bluetooth-connected total station was used to record and calculate the observation results. The validity of the method was verified by comparing the relative amount of change in displacement from a certain point in time.

Based on the characteristics of incremental launching construction, both the forward pushing and lateral alignment corrections are performed while the girder is lifted. The entire launching process can be divided into two conditions: lifting the girder and lowering the girder. This paper focuses solely on the lifting condition. The position of the girder after the preliminary lifting is taken as the reference to establish an initial pixel coordinate system for analysis.

(1) Under the lifting beam condition, Figure 19 shows the displacement measurement results of the two methods in the incremental launching process. The NRMSE value of lateral deflection is calculated to be 3.09%, and the NRMSE value of the forward displacement of the incremental launching advancement is 1.83%. Due to the complex field environment, which includes numerous disturbing factors, the system error increases significantly. However, the trend of the monitoring results from this method is largely consistent with that from the total station, indicating better monitoring results overall.

The results of the visual monitoring section are shown in Figure 20, which contains the measurements of one incremental launching stroke. Specifically, at about 9:34, the previous incremental launching stroke ends, and the jacking equipment and the girder descends, at which time a clear phase of decreasing displacement can be observed, as shown in subfigure A, as well as a corresponding change in deflection, as shown in subfigure C. The subsequent flat section indicates a phase of construction preparation and waiting. At about 9:42, a new incremental launching stroke starts, and the beam begins to jack up, at which time a clear phase of displacement increase can be observed, as shown in subfigure B, as well as a corresponding change in the deflection, as shown in subfigure D. The above displacement changes are due to the change in the shooting distance between the camera and the bottom of the beam when the beam is dropped and lifted, resulting in a change in the position of the target in the field of view. The trip started jacking at about 9:46 and ended at about 9:50 with the girder drop. Comparing the displacements of the two adjacent jacking positions, the difference in forward displacement is 1.6 mm and the difference in lateral offset is 0.82 mm, which can indirectly illustrate the effectiveness of this method.

(2) Under the lowering beam condition, considering the change in the shooting distance when the beam is falling, the scale factor and the position of the same target in the pixel coordinate system are changed, and it is not possible to use the previously set pixel coordinate system for calculation, which can be used as a direction for future research. In order to analyze the displacement changes under the two working conditions of falling and rising beams, two initial pixel coordinate systems can be established: one for the case where the shooting distance is the smallest when the beam is falling, and the other for the case where the shooting distance is the largest when the beam is jacked up. The former is used to analyze the displacement change after the beam is raised, and the latter is used to analyze the displacement change after the beam is lowered.

The feasibility and effectiveness of the proposed method were verified by a combination of simulation tests and real bridge tests.

## 5. Conclusions

This paper focuses on the interdisciplinary application of civil engineering and computer science, with our team primarily oriented toward using computer vision to address practical challenges in civil engineering. Our team has also combined drones and monocular vision for the intelligent detection of bridge bolts [35] and is conducting research on utilizing stereo machine vision for road damage assessment and stereo vision measurement. We hope that our study on using computer vision in underwater construction processes will provide valuable insight into the broader application of computer technology in other areas of civil engineering. In this paper, we propose to combine the YOLOv5 target detection algorithm with the DeepSORT multi-target tracking algorithm for monitoring displacements during bridge incremental launching construction. This approach enables real-time monitoring of both transverse offsets and incremental forward displacements in scenarios involving straight bridges with a linear bottom surface. The key points are summarized as follows:

(1)The feasibility of the proposed method for real-time monitoring of bridge launching construction has been demonstrated through simulation tests and real bridge verification, confirming the reliability of the detection accuracy. By integrating YOLOv5 and DeepSORT with geometric matching, it enables the monitoring of forward displacement and lateral deflection of the bridge launching process with a straight bottom surface line shape. Since the proposed method can measure both longitudinal and lateral displacements at the bridge’s base, its application is feasible for curved bridges or skewed bridges.(2)In this paper, we construct a special target dataset for bridge launching scenarios and further process the image features of the target ROI. We combine the algorithms of edge contour line extraction, convex packet detection, and K-means clustering to achieve the localization of geometric centroid of the crosshatched target. YOLOv5n was applied for target detection on the test dataset and achieved the best performance, with an mAP of 91%, a precision of 99.2%, and a recall of 94.2%.(3)In the simulation test, the visual measurement results are compared with the total station measurement results, with the maximum NRMSE of the forward displacement being 0.545% and the maximum NRMSE of the lateral offset being 1.73%. In the real bridge test, the initial pixel coordinate system was established with the girder launching position to compare the measurement results of the launching displacement during launching, in which the NRMSE of the jacking forward displacement was 1.83%, while the NRMSE of the lateral offset was 3.09%. Due to the site having more disturbing factors, the system error increases obviously, but the trend of the monitoring results of this method is basically consistent with the total station, which has better monitoring results.

## Figures and Tables

**Figure 1 sensors-24-07385-f001:**
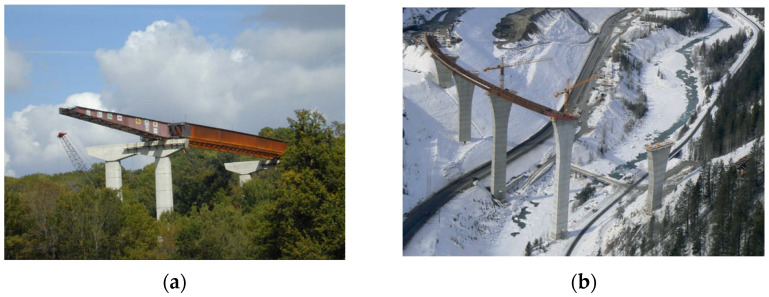
(**a**) Launching of Iowa River Bridge Pier 4; (**b**) launching of the first girder pair for the Park Bridge.

**Figure 2 sensors-24-07385-f002:**
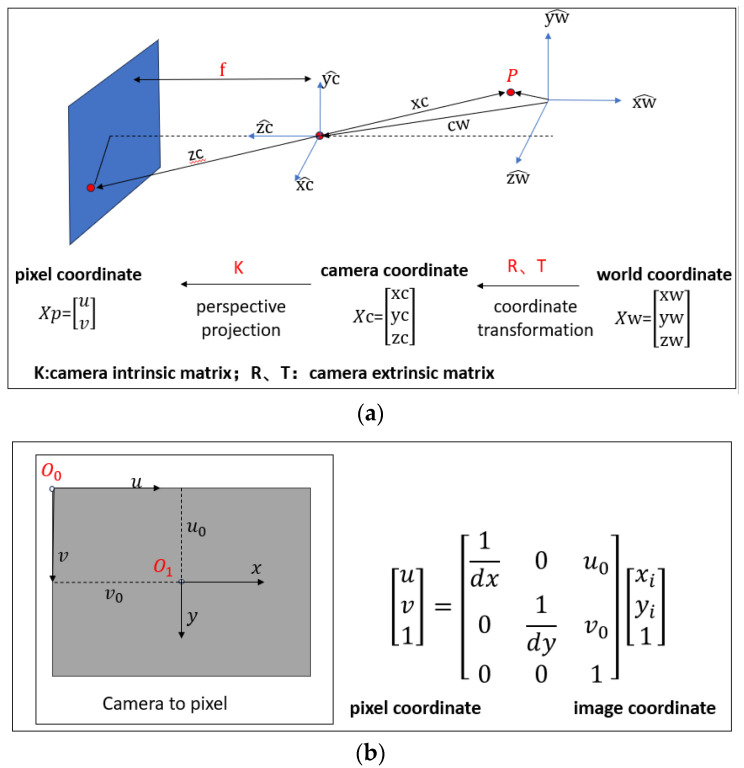
(**a**) Camera linear model; (**b**) image coordinate to pixel coordinate (3D-2D).

**Figure 3 sensors-24-07385-f003:**
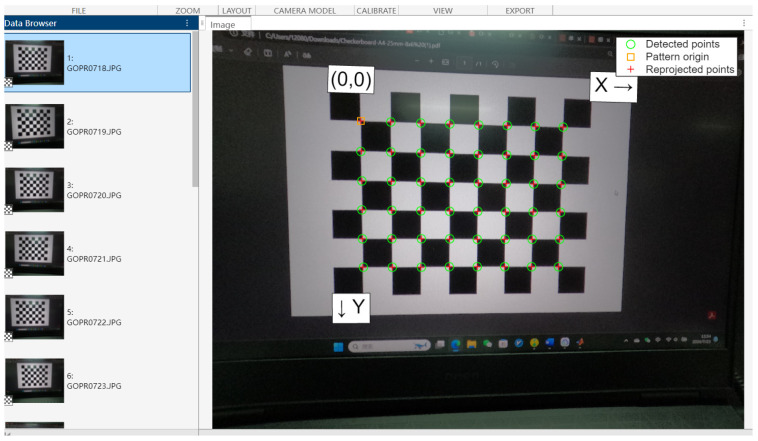
Camera calibration by MATLAB 2024.

**Figure 4 sensors-24-07385-f004:**
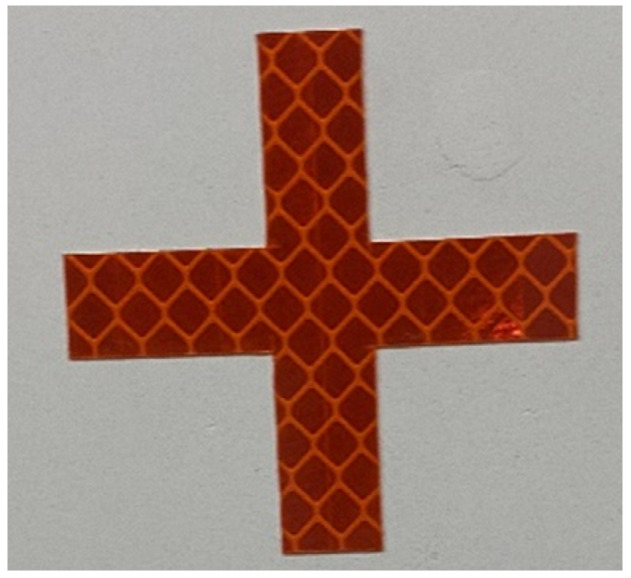
Cross target.

**Figure 5 sensors-24-07385-f005:**
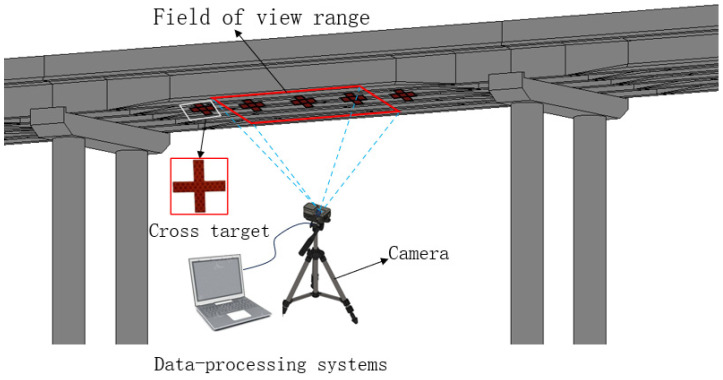
Schematic diagram of visual measurement system.

**Figure 6 sensors-24-07385-f006:**
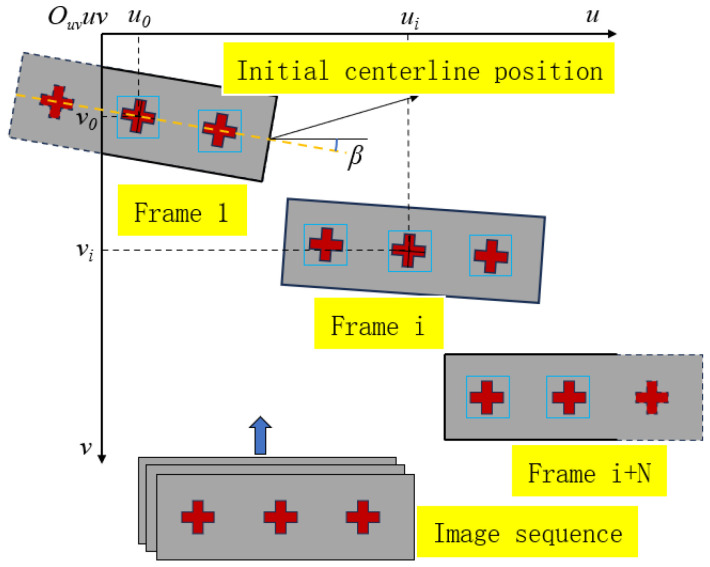
Derivation of the initial position relation of the target.

**Figure 7 sensors-24-07385-f007:**
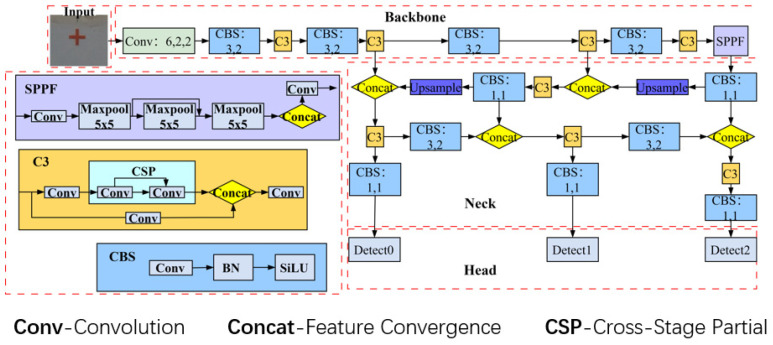
Network structure of YOLOv5s.

**Figure 8 sensors-24-07385-f008:**
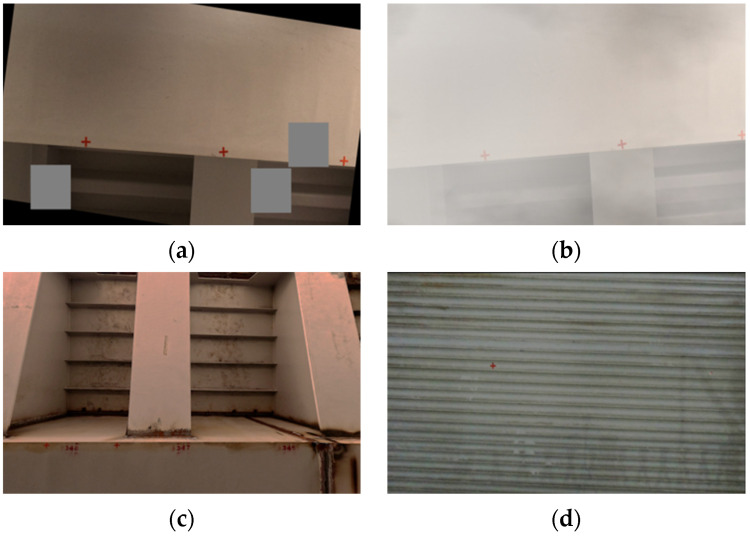
Partial data enhancement effect diagram: (**a**) Cutout; (**b**) Synthetic Fog Enhancement; (**c**) Luminance; (**d**) Motion Blur.

**Figure 9 sensors-24-07385-f009:**
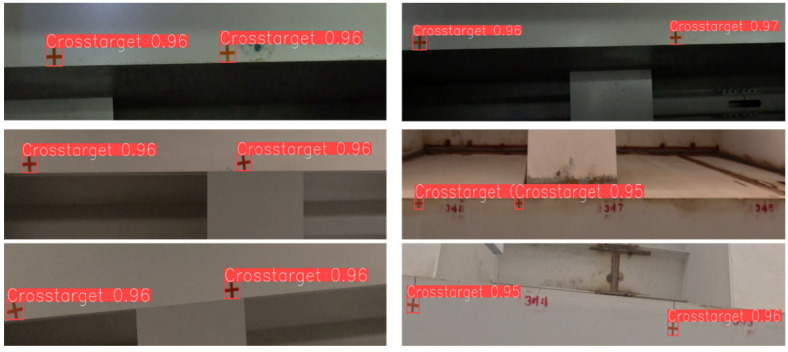
YOLOv5 target detection model positioning effect diagram.

**Figure 10 sensors-24-07385-f010:**
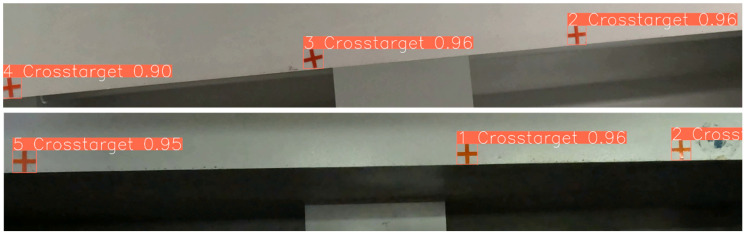
Cross target tracking results.

**Figure 11 sensors-24-07385-f011:**
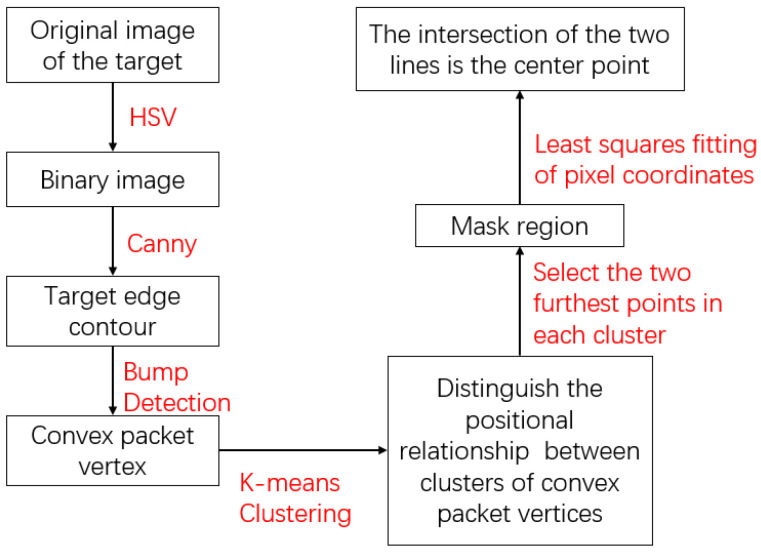
Center-point calculation process.

**Figure 12 sensors-24-07385-f012:**
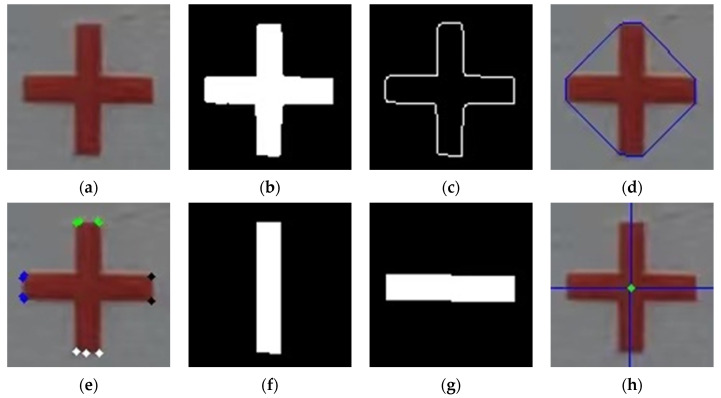
Step-by-step effect of the center-point solution process: (**a**) cross target; (**b**) binary image; (**c**) edge contours; (**d**) bump detection; (**e**) convex packet vertex clustering; (**f**) Region 1; (**g**) Region 2; (**h**) straight-line fitting for intersection.

**Figure 13 sensors-24-07385-f013:**
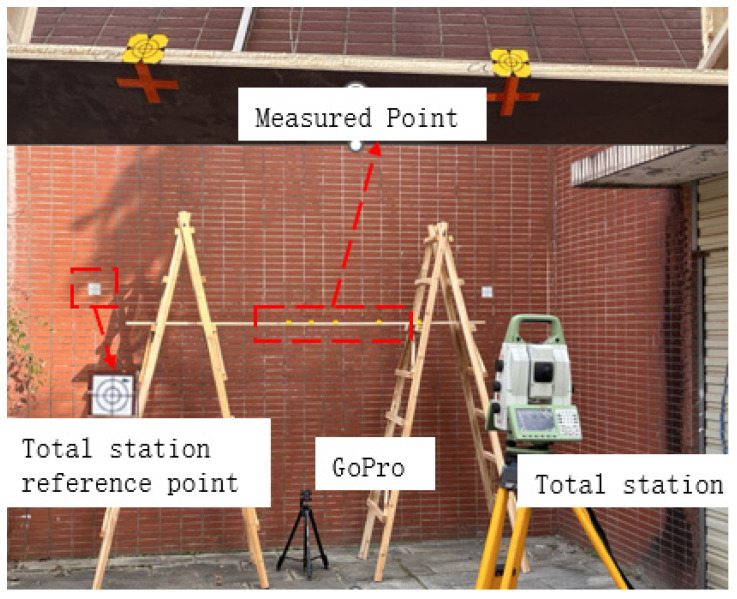
Plank simulation test setup.

**Figure 14 sensors-24-07385-f014:**
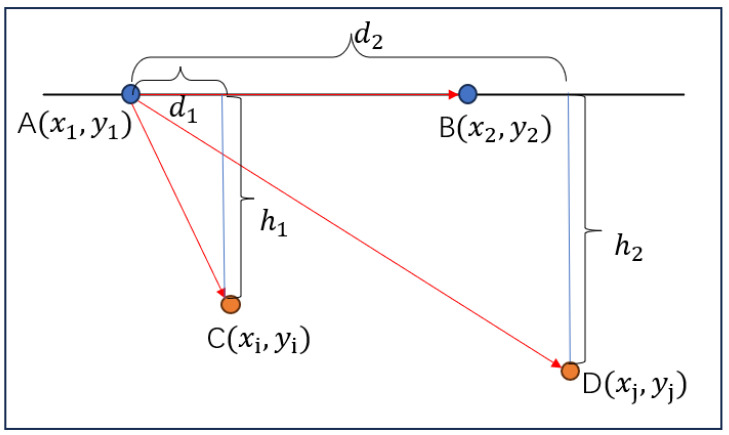
Schematic diagram of the lateral offset and forward displacement of the measured point.

**Figure 15 sensors-24-07385-f015:**
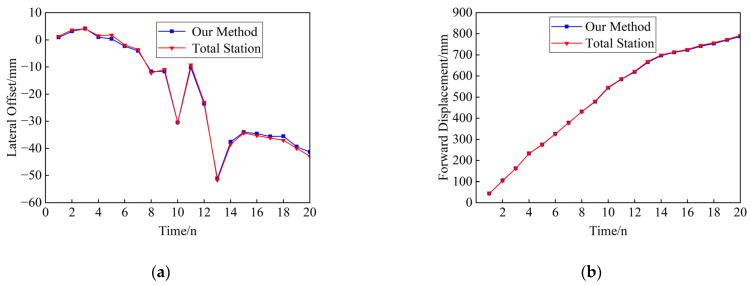
Comparison of displacement measurement results for measurement point 1: (**a**) comparison of lateral offset measurements at measurement point 1; (**b**) comparison of incremental launching forward displacement measurements at measurement point 1.

**Figure 16 sensors-24-07385-f016:**
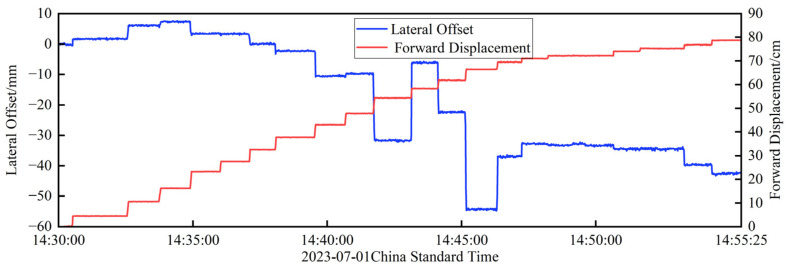
Visual measurement of displacement results.

**Figure 17 sensors-24-07385-f017:**
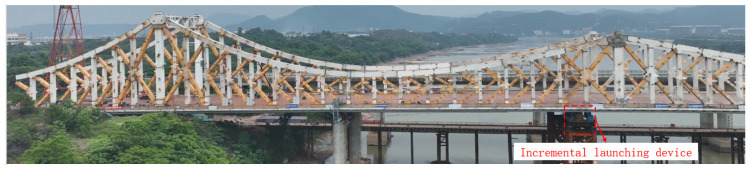
Lixizhou Bridge.

**Figure 18 sensors-24-07385-f018:**
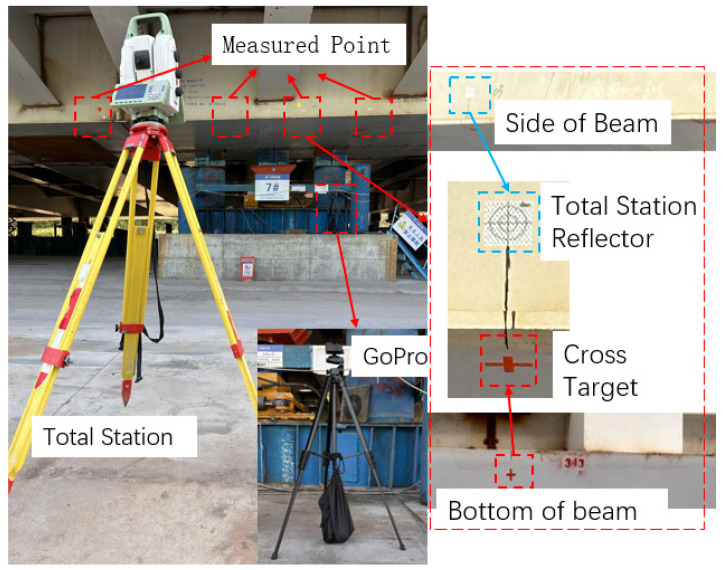
Field test layout.

**Figure 19 sensors-24-07385-f019:**
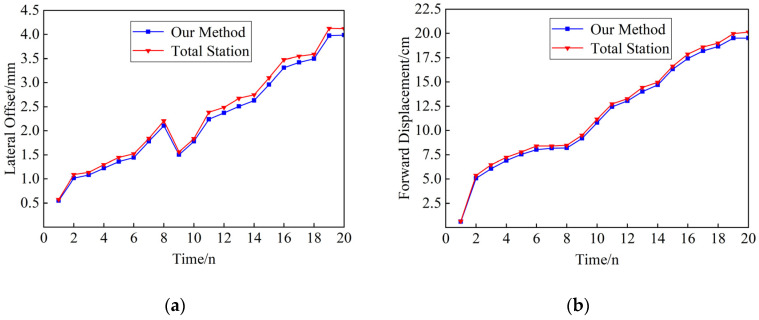
Comparison of incremental launching displacement measurements: (**a**) comparison of lateral offset measurements; (**b**) comparison of incremental launching forward displacement measurements.

**Figure 20 sensors-24-07385-f020:**
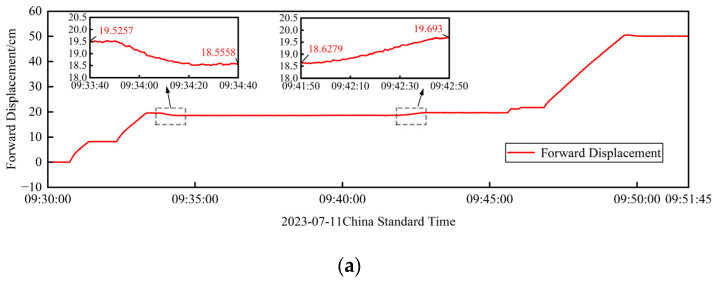
Visual measurement results for a time period: (**a**) forward displacement visual measurements; (**b**) results of the visual measurement of the lateral offset.

**Table 1 sensors-24-07385-t001:** Comparison of target detection results of four models of YOLOv5 series.

Models	Accuracy/%	Recall Rate/%	mAP/%	Quantity	Speed/ms	GPU
YOLOv5s	98.7	95.2	92	7.01 × 10^6^	34.7	NVIDIA GeForce RTX 4060 by NVIDIA in Santa Clara, California
YOLOv5s6	99.1	96.9	94.2	1.23 × 10^7^	103.2	
YOLOv5n	99.2	94.2	91	1.76 × 10^6^	13.3	
YOLOv5n6	99.2	95.2	93.2	3.09 × 10^6^	40.5	

**Table 2 sensors-24-07385-t002:** Results of the first five displacements at point 1.

Time	Total Station		Our Method	
	Lateral Offset/(mm)	Forward Displacement/(mm)	Lateral Offset/(mm)	Forward Displacement/(mm)
0	0.91379	43.9803	43.99269	1.28755
1	3.10069	105.511	104.97863	3.6862
2	4.19259	162.935	162.97744	4.06527
3	0.96848	233.14	233.9952	1.52938
4	0.36514	275.647	275.99434	1.80388

**Table 3 sensors-24-07385-t003:** Analysis of simulation test error results.

Points	NRMSE/%	
	Forward Displacement	Lateral Offset
1	1.42	0.349
2	1.73	0.545
3	1.49	0.512
4	1.55	0.455

## Data Availability

No new data were created or analyzed in this study. Data sharing is not applicable to this article.

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
