# Peer review of "Machine Vision-Based Real-Time Monitoring of Bridge Incremental Launching Method"

_sensors, 2024, doi:10.3390/s24227385_

Round 1

Reviewer 1 Report

Comments and Suggestions for Authors

Dear authors,

Firstly, I would like to thank you for giving me the opportunity to review your research paper. I have carefully read the MDPI Sensors paper titled " Machine vision-based Real-time Monitoring Of Bridge Incremental Launching Method ".

Comments and Suggestions for Authors:

·         The obtained results (Section 4: Experimental Validation and Results) are poorly described in the research part (e.g. Figure 14, 15, 18, 19 and table 2.). A detailed analysis of the results is required.

·         The conclusions are very short, they need to be supplemented. I recommend including the obtained results, including numerical values.

  I would recommend Accept article after minor revision at MDPI Sensors.

Good luck on Your future research.

Reviewer 2 Report

Comments and Suggestions for Authors

The paper demonstrates the feasibility of using machine vision for real-time displacement monitoring during bridge incremental launching. The proposed method combines YOLOv5 for target detection and DeepSORT for multi-target tracking, showing good agreement with total station measurements in both simulation and real-world tests. However, the following comments should be well addressed before the paper can be considered for publication.

(1) The method is only validated for straight bridges with straight bottom surfaces. Its applicability to curved or more complex bridge structures is not addressed.

(2) While NRMSE values are provided, there's no in-depth discussion of potential error sources or the system's limitations under different conditions.

(3) The paper doesn't thoroughly address how changes in camera position or orientation during the launching process might affect measurement accuracy.

(4) The impact of varying lighting conditions, weather, or vibrations on the system's performance is not extensively explored.

(5) The paper doesn't discuss the long-term reliability or durability of the proposed system in harsh construction environments.

(6) While detection speeds are mentioned, the overall real-time processing capability of the entire system is not clearly stated.

(7) The paper doesn't address how the system might be scaled up for larger bridges or multiple monitoring points.

(8) The figures are generally informative, but some could benefit from higher resolution and clearer labeling, especially Figures 7-9.

(9) Table 1 provides a useful comparison of YOLOv5 models but could be enhanced with a brief discussion of the tradeoffs between accuracy and speed.

Comments on the Quality of English Language

Overall, the English language quality in this paper is good, However, there are some minor areas where improvements could be made: Some sentences are overly long and complex, which can make them difficult to follow. Breaking these into shorter, more focused sentences would improve readability.

Reviewer 3 Report

Comments and Suggestions for Authors

The paper deals with a methodology for the structural monitoring of bridges, within the construction context of incremental launching method.

Some interesting features, among the paper aims, can be highlighted, particularly, for real-time monitoring, vision-based monitoring (non-contact technique) and machine automated monitoring.

Despite such features and the possible interest toward the subject, for the engineering community, several observations may be raised with respect to the manuscript, as in the following:

- the Introduction section is rather limited, with respect to the state-of-the-art in bridge structural monitoring, structural diagnosis and combined innovative methods for automated approaches, also within the framework of Inverse Analysis;

- the core sections of the paper adopt two already developed algorithms, from the literature, with significant lack of explanations and without innovative contributions;

- the main contents are limited to a practical study application, therefore with limited scientific contributions and limited elaborations, and discussion, on the gathered results;

- the Conclusions section is displayed to be too brief and generally missing comments, observations, interpretations, highlights, etc.

Comments on the Quality of English Language

The paper text appears to be too concise and frequently not explanatory. From the Abstract to the Conclusions, several sentences turn out to be understandable with difficulties, due to the lack of details, explanations, descriptions and correlations.

Round 2

Reviewer 2 Report

Comments and Suggestions for Authors

This paper can be accepted  in the current version.

Reviewer 3 Report

Comments and Suggestions for Authors

The currently revised version of the paper does not significantly accounts for the reviewers’ comments, since the implemented modifications provide minor changes, without effective improvements in none of the core sections (literature survey, innovative methodology development, application elaborations, result interpretations).

Comments on the Quality of English Language

The paper text frequently appears not sufficiently clear nor explanatory.

Please, also avoid non-English texts (such as in the rebuttal letter).

Round 3

Reviewer 3 Report

Comments and Suggestions for Authors

The currently revised version of the manuscript significantly improves the discussed content, although the innovative contributions still remain rather limited.

Further comments are related to:

- the Introduction section, still missing of some contributions related to computational approaches by Inverse Analysis (see, e.g., https://doi.org/10.3390/ma16247512);

- proper aims and limitations of the work, as commented by the Authors in the cover letter (response 3), to be also specified within the paper contents.

Comments on the Quality of English Language

Please, carefully revise the manuscript to polish the text and remove misprints.
